

# Avian biodiversity in central California vineyards

Lindsay Peria and Clinton D. Francis

Biological Sciences Department, California Polytechnic State University-San Luis Obispo, San Luis Obispo, California, United States

## ABSTRACT

Avian biodiversity is declining globally, and conservation lands alone will likely not be able to support vibrant avian communities long-term. However, the integration of wildlife friendly practices into agricultural lands could support many birds that have lost habitat to agricultural and urban development. Here, we assessed how structural, natural, and anthropic vineyard characteristics in Edna Valley, California influence avian species occupancy, taxonomic diversity, and functional diversity from 31 point counts that, collectively, captured large gradients in environmental variation that exist in and around Edna Valley vineyards. Although we found the occupancy of relatively few species to covary with structural, natural, and anthropic vineyard characteristics, increasing canopy cover was associated with higher taxonomic and functional bird diversity in vineyards, and canopy cover and sound level were associated with shifts in bird community composition. We also found some evidence that proximity to surface water and agricultural cover variation surrounding vineyards could increase functional bird diversity; however this requires further investigation. Additionally, vineyard cover was negatively associated with functional evenness, but we did not find evidence that it was related to any other taxonomic or functional diversity metrics. This research could help guide wildlife friendly vineyard management throughout California to help increase the number of birds that can utilize these modified lands for foraging, shelter, and to connect larger areas of protected land.

## INTRODUCTION

Global wildlife diversity is declining due to urbanization, climate change, and additional environmental threats, and despite large efforts to combat these declines, the threats show no significant signs of slowing down (*Butchart et al., 2010*; *Newbold et al., 2015*). Birds have not been immune to these changes with an estimate of almost 3 billion birds lost in the U.S. and Canada since 1970 (*Rosenberg et al., 2019*). Habitat loss and pesticide exposure are major causes of bird population decline in agricultural areas, and 74% of farmland associated species have declined in recent years (*Stanton, Morrissey & Clark, 2018*), and mediterranean areas in particular, including parts of California, have a high rate of conversion of natural habitats to specialty crops, such as wine grapes (*Viers et al., 2013*). Agriculture is a major source of revenue in California, and vineyards cover almost 300,000

Corresponding author
Lindsay Peria,
lindsayp773@gmail.com

hectares of land in the state (*California Department of Food and Agriculture & USDA National Agriculture Statistics Service, 2024*). San Luis Obispo County, located on the Central Coast of California, is a local hub for viticulture, the study of cultivating grapes including management of vineyards, where wine grapes are the most profitable crop valued at over $300,000,000 in 2023 and spanning over 18,000 hectares (*County of San Luis Obispo, 2024*). In regions where large amounts of land are already converted to agriculture, and land sparing for wildlife use is not an option, there are increasing calls for agricultural lands to apply land sharing, managing the land with a wildlife conservation focus, to preserve biodiversity and the ecosystem functions and services that come with higher biodiversity (*Kremen & Merenlender, 2018*).

In addition to the habitats that vineyards can provide for wildlife, vineyards can also benefit greatly from ecosystem services that birds provide (*Viers et al., 2013*). Birds can reduce crop damage from insects such as caterpillars (*Bereczki et al., 2014*; *Mols & Visser, 2002*), and diverse avian communities provide more agricultural insect control (*Barbaro et al., 2017*). Mammalian vineyard pests such as voles, gophers, and mice can also be controlled by birds (*St. George & Johnson, 2021*). Unfortunately, many agricultural areas are simplified to the point that they are not suitable for birds, diminishing the valuable ecosystem services that they can provide on working lands (*Rusch et al., 2016*). As they are commonly managed, bird abundance is generally lower, and the communities are less diverse in vineyards compared to surrounding habitat (*Jedlicka, Greenberg & Raimondi, 2014*; *Pithon et al., 2016*). However, when resources such as shelter, food, and nesting materials are interspersed within or surrounding agricultural areas, birds utilize the areas (*Muñoz-Sáez, Kitzes & Merenlender, 2021*).

In the past decade, there has been a growing interest in understanding which viticulture practices increase avian diversity in vineyards (*Barbaro et al., 2021*; *Belkhiri et al., 2023b*; *Assandri et al., 2017*), and some studies have focused on larger landscape scale factors that influence bird communities in vineyards (*Assandri et al., 2016*; *Belkhiri et al., 2023a*; *Bennett et al., 2014*). Notwithstanding the importance of these efforts, much of this research occurred in Europe, northern Africa, and Australia, thus their findings may not be relevant to the avifauna and ecosystems affected by the expanding viticulture industry in California. For instance, Europe has a much longer record of intense land use than most of California. This long history has likely reduced and homogenized the available pool of species in contemporary studies focused on responses to viticulture practices. Because California's avifauna does not share this history of coexistence with intense agriculture, community members may respond quite differently to these novel landscapes. Finally, most conservation management practices in the United States of America are species-focused, thus knowledge of species-specific responses to environmental features are often required for effective conservation efforts.

It is also important to understand both taxonomic and functional diversity in avian communities to inform conservation efforts and the variety of habitat utilization and resource use within a community (*Moore & Brodie, 2024*). Taxonomic diversity reflects the number and/or relative abundance of species and functional diversity typically reflects variation in functional traits that determine how organisms interact with the environment.

In vineyard settings, increasing avian taxonomic diversity could positively influence psychological ecosystem services provided to vineyard workers and visitors (*Ferraro et al., 2020*; *Methorst, 2024*) whereas increases in functional diversity can have a myriad of ecosystem benefits, including services directly beneficial to vineyard management, such as pest control (*Jedlicka, Greenberg & Letourneau, 2011*; *St. George & Johnson, 2021*).

This study aims to help inform bird-friendly vineyard management in the central coast and elsewhere in California. Specifically, building upon previous research in California vineyards which largely focused on nesting, ecosystem services, and inter-species interactions (*e.g.*, *Fiehler, Tietje & Fields, 2006*; *Jedlicka, Greenberg & Letourneau, 2011*; *Jedlicka, Greenberg & Raimondi, 2014*; *Muñoz-Sáez et al., 2020*). We focus on identifying the natural, anthropic, and structural characteristics of viticultural landscapes that are associated with higher taxonomic and functional avian diversity. We predict that the occupancy of forest and shrubland associated birds would increase with canopy cover, canopy height variation, and shrubland cover, grassland bird occupancy would be positively associated with grassland cover and negatively associated with canopy cover, and that all birds except agricultural specialists would be negatively associated with agricultural cover and sound level. We also predict that avian community diversity would be higher in areas with high canopy cover, canopy height variation, grassland cover, shrubland cover, and closer to water, and diversity would be lower in areas with high agricultural cover and high sound levels.

## METHODS

We conducted our study in an approximately 9,000-hectare study area in Edna Valley, San Luis Obispo County, California (Fig. 1). The area is dominated by vineyards with other agriculture such as orange groves and strawberry fields, natural grasslands and shrublands, and exurban residential development. To survey bird communities in and around the vineyards, we chose 25 point count locations immediately adjacent to vineyards that captured variation in natural and human modified land cover characteristics. We also chose six sites that were surrounded by natural grassland and shrubland, with no vineyard cover and over 70% woodland, shrubland, and grassland cover within 150 m to serve as locations with minimal anthropic influences. Collectively, the point count locations were surrounded by considerable variation in land cover (Fig. S1). All point count locations were at least 400 m from one another to ensure independence of counts (*Ralph, Droege & Sauer, 1995*). All field work occurred from public right-of-ways and counting birds detectable from these locations did not require approval from any entity.

### Point counts

To quantify the avian community among Edna Valley Vineyards, we visited each point count location three times between April 21 and June 10, 2024. On each visit, we conducted two back-to-back 5-min point counts, thus totaling 6 point counts per location. All surveys were performed between 6:00 am and 10:15 am to capture high and stable avian activity (*Ralph, Droege & Sauer, 1995*), and surveys were done by the same person to eliminate differences in inter-observer bias. During these point counts, we kept track of

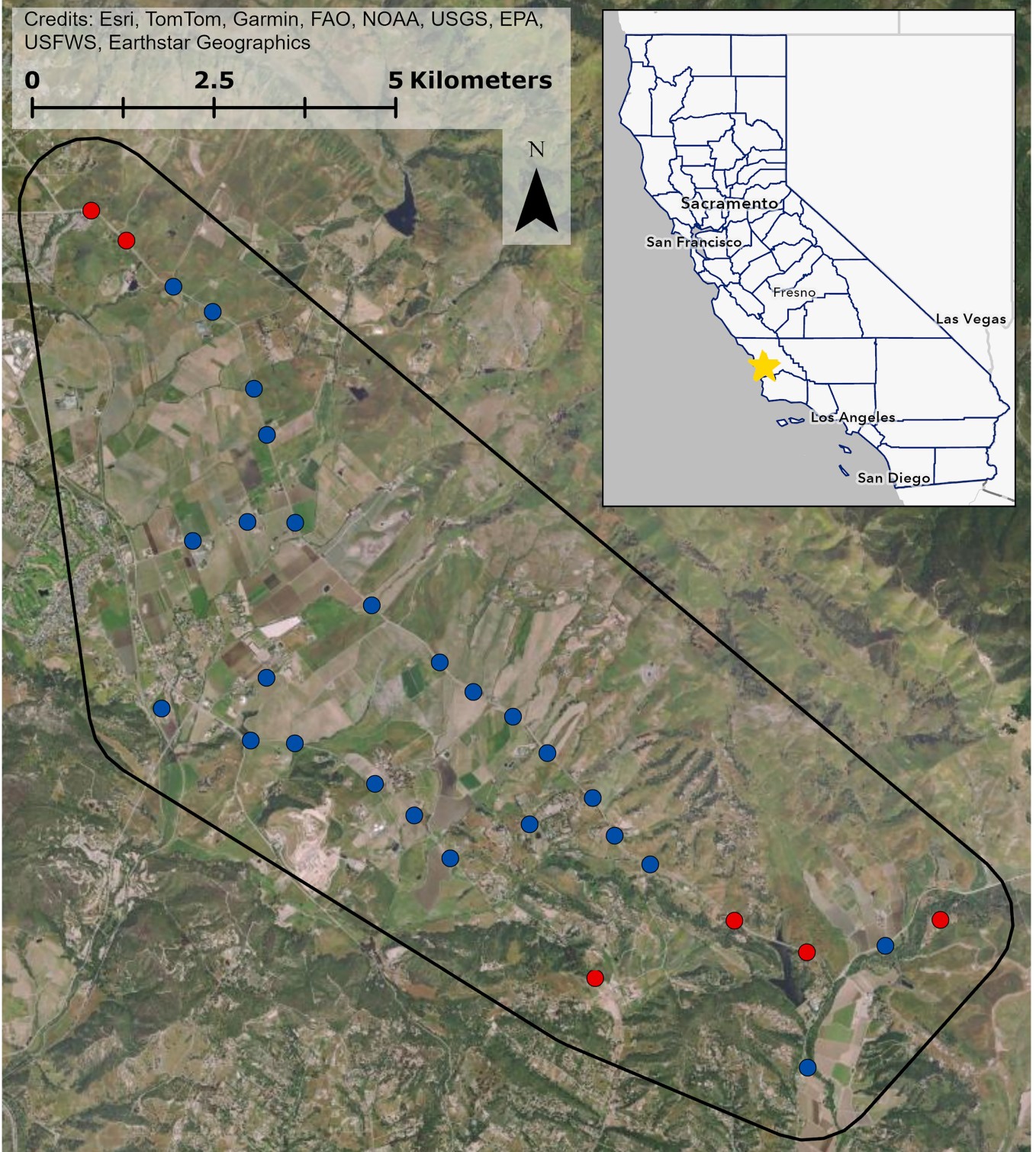

**Figure 1 Study site in Edna Valley, CA.** Vineyard sites are in blue, control sites are in red, and the extent of the study site is in black. Inset shows California counties outlined in blue, and the study site location on the central coast of California is marked with the star. Map image is the intellectual property of Esri and is used herein under license. Copyright © 2025 Esri and its licensors. All rights reserved.

individuals to only count individual birds once per survey. We recorded the distance, bearing, behavior, activity, and detection method for the first detection of each individual as either aural or visual. For subsequent analyses, observations were limited to those within 100 or 150 m to ensure that we had a reasonable probability of observing any individuals present. Detection probabilities were very similar for both a 100 m and 150 m radius (Table S1). Most species observations were limited to within 100 m, but we used a larger 150 m radius for California scrub-jay (*Aphelocoma californica*), acorn woodpecker (*Melanerpes formicivorus*), Nuttall's woodpecker (*Dryobates nuttallii*), red-tailed hawk (*Buteo jamaicensis*), red-shouldered hawk (*Buteo lineatus*), mourning dove (*Zenaida macroura*), Eurasian collared-dove (*Streptopelia decaocto*), California quail (*Callipepla californica*), and American crow (*Corvus brachyrhynchos*) observations because they are larger, more conspicuous species that are easy to detect from farther away.

## Noise levels

Because human and natural sources of background noise have been shown to influence the distribution of individual bird species and the structure of avian communities (*Gomes et al., 2021*; *Bayne, Habib & Boutin, 2008*; *Francis, Ortega & Cruz, 2009*) and also influence detection probabilities during point counts (*Ortega & Francis, 2012*; *Pacifici, Simons & Pollock, 2008*), we quantified background sound levels in two ways. First, to characterize sound levels typical of each point count location, an AudioMoth (1.10.0) was placed on fences, utility poles, trees, and other structures within 4 m of the point count location and at a height of approximately 1 m to record sounds for 24 h. We completed each recording on a weekday to ensure differences in traffic volume were not due to differences in weekday and weekend traffic. The AudioMoths were placed in a plastic bag and a small pouch made of faux fur that functioned as a custom windscreen. AudioMoths were set to record on medium gain with a 48 kHz sample rate (16 bit). We used end-to-end calibration to calibrate each AudioMoth with a LarsonDavis 824 Sound Pressure Meter. From the 24-h recordings we calculated time-averaged A-weighted decibels ($L_{eq}$, re 20 µPa) using Kaleidoscope Pro 5.6.8. In preliminary analyses we explored whether morning sound levels (5 am-11 am) were better than the full 24-h recordings in describing diversity patterns, but Akaike Information Criterion corrected for small sample size ($AIC_c$) comparisons suggested that sound levels based on this truncated window were equivalent or less useful than sound levels derived from full recordings. Second, to account for the possible influence of background sound levels on detections during point counts, we measured the time-averaged A-weighted decibels ($L_{eq}$, re 20 µPa) of average background noise with a MicW i436 omni directional microphone paired with the SPLnFFT Sound Meter iPhone application (v7.1) during each survey. Although audio recorders could potentially impact the privacy of people by inadvertently recording them, pedestrian traffic is exceedingly uncommon at all recording points. Pedestrians were never encountered during any of the surveys or other field work. Furthermore, the audio recordings were only used to quantify sound levels at each site by audio processing software, therefore any comments made by a person that was captured by recorder would never be heard by any person.

## Structural and land cover variables

Structural and land cover variables were calculated for the area within 150 m of each site. Land cover was hand digitized based on satellite imagery provided by *Esri et al. (2024)* and from in person observations in order to ensure that any recent land cover alterations were included. Land was categorized as grassland, shrubland, woodland, vineyard, orchard, row crop (*e.g.*, oats, strawberries, or vegetables), or developed. The 31 survey sites did not have adequate variation in woodland cover surrounding them, so this variable was excluded from subsequent analyses. Surrounding vineyard cover varied from natural reference sites with 0% vineyard cover to sites with about 80% vineyard cover.

We used Light Detection and Ranging (LiDAR) data from USGS 3D Elevation Program (*U.S. Geological Survey, 2021*) to calculate percent canopy cover, the percent of an area covered by trees over 4 m tall, and variation in canopy height with the 'lidR' 4.1.2 package in R (*Roussel et al., 2020*; *Roussel & Auty, 2024*). We used the *rasterize_canopy* function to create a raster of the canopy height within a 150 m radius of each survey point and calculated the percent canopy cover as the percent of canopy over 4 m tall within each site. We limited canopy to that over 4 m to exclude grape vines and fruit trees in orchards as they are accounted for in the vineyard and orchard percent land cover variables; however, this variable captures canopy cover from naturally occurring trees and ornamental trees. Variation in canopy height was calculated as the standard deviation of the canopy raster for each site.

Distance to surface water was calculated as the distance from the survey point to the nearest stream or other surface water based on a USGS national hydrography dataset with manual corrections for human-made ponds that are not included in the dataset, but observed during site visits or on satellite imagery (*U.S. Geological Survey, 2019*).

## Data analysis

All analyses were conducted in R version 4.4.1 (*R Core Team, 2024*). To determine which, if any, environmental variables were correlated with each community metric that we describe below, we fit three models based on our original hypotheses that (1) structural complexity, (2) anthropic environmental features, and (3) natural land cover influenced bird communities at vineyards. Structural complexity models included percent canopy cover, canopy height variation, and distance to surface water. Anthropic environmental feature models included vineyard cover, row crop cover, orchard cover, developed land cover, and noise level. Natural land cover models included grassland and shrubland cover. Model fit was assessed by checking normality of residuals, normality of random effects, multicollinearity, and homogeneity of variance using the *check_model* function in the "performance" R package (*Lüdecke et al., 2021*). We ranked each model from the three hypothesis categories based on $AIC_c$. We chose this modeling technique as opposed to a global model to avoid overfitting the model with too many variables and a relatively small sample size. We then assessed the strength and precision of estimated effects in the top model of each hypothesis category using effect sizes and error metrics, plus *p*- and F-values. Predictor variables from each hypothesis category that were considered to have an apparent effect on the response were then incorporated into a *post hoc* model that

combined influential variables from each hypothesis category. Interpretation of the influence of predictor variables on responses were based on this final *post hoc* model. Following recent advice from statisticians (*Halsey, 2019*; *Hurlbert, Levine & Utts, 2019*; *Wasserstein, Schirm & Lazar, 2019*), in this study we sought to use a more nuanced approach to reporting responses among the avian community to landcover variation than the dichotomous approach of significance testing. As such, for linear models we report estimated effect sizes and standard errors (SE) alongside t/z-values and *p*-values to help convey the magnitude and precision of estimated effects. For analyses involving beta-diversity we complement the presentation of F- and *p*-values with $R^2$-values to provide a more rounded understanding of which predictor variables explain the most variation in community structure. Thus, we do not consider any effects "significant," but instead present all apparent trends and attempt to contextualize the magnitude and precision of estimated effects in the text.

### Species occupancy

To determine the probability of occupancy for species that were detected in this study, we created a survey detection non-detection matrix for each species that was detected within 100 m, or 150 m for larger species listed above.

We used the *occu* function in the *unmarked* 1.4.3 package in R to estimate the species' probability of occupancy for the 22 most common species with sufficient detection histories based on model diagnostics (*Fiske & Chandler, 2011*). Sound level during the survey, day of the year, and time of day were included in the models as detection covariates. We assessed three different occupancy models based on our hypotheses that anthropic characteristics, natural characteristics, and structural complexity of vineyards would impact species occupancy, and we selected the most informative occupancy covariates based on $AIC_c$ and *p*-value to include in a final *post hoc* model. Goodness of fit for these models was assessed using Mackenzie and Bailey goodness-of-fit test.

### Taxonomic community models

It is likely that the same individuals were counted in more than one point count at the same site, so the cumulative number of individuals for each species should not be used for a community matrix for each site. Therefore, the maximum number of individuals per species detected during a single point count survey was used to calculate a community matrix for each site. Based on species rarefaction curves, the number of species observed did not approach the actual number of species at each site (Fig. S2), thus we chose to use coverage, the estimated degree to which the survey accurately captures the avian community (*Roswell, Dushoff & Winfree, 2021*), to better estimate diversity metrics in the taxonomic community analysis. We used the *iNEXT* function from the 'iNEXT' 3.0.1 package (*Hsieh, Ma & Chao, 2016*) to estimate coverage for each site. We then used the lowest coverage from any site to estimate the coverage-based Shannon index and species richness with the *estimateD* function. We chose these metrics because of their contrasting weights given to rare or abundant species; richness giving more weight to rare species and the Shannon index giving more weight to relative abundance (*Roswell, Dushoff & Winfree,*

*2021*). We used linear models to determine which, if any environmental variables were correlated with coverage-based Shannon and species richness estimates at these sites.

Based on preliminary data exploration, the relationship between canopy cover and the coverage-based estimates of evenness and richness did not appear to be linear. Therefore we compared linear, second degree polynomial, and third degree polynomial models to determine which best reflected the relationship between community diversity metrics and canopy cover. Based on $AIC_c$, we concluded that the second degree polynomial of canopy cover received the most support and included it in subsequent analyses.

We also fit a PERMANOVA with the *adonis2* function in the 'vegan' version 6.1 package (*Dixon, 2003*) to investigate turnover between the sites. We used the *rankindex* function, also from the 'vegan' package, to determine which beta diversity index, Euclidean, Manhattan, Gower, Bray-Curtis, or Kulczynski, best captured variation in community composition. The Gower Index best captured variation in community structure and was used as the response variable in the PERMANOVA models. As with the species occupancy and community diversity models, three models were made initially based on structural complexity, natural cover, and agricultural cover variables, and F and *p*-values were used to determine the most informative variables. Those variables were then added to a *post hoc* model that best explains beta diversity. We used non-metric multidimensional scaling plots to visualize the PERMANOVA and used a stress test to determine that three dimensions best reflect the variation among the communities (stress = 0.156).

### Functional community models

To assess the functional diversity of the bird communities, we used the *dbFD* function from the 'FD' version 12.3 R package to evaluate functional richness, functional evenness, and functional dispersion (*Laliberte, Legendre & Shipley, 2014*). Each of these metrics is based on the species' location in multidimensional trait space, in which each axis represents a different morphological trait. Functional richness is calculated as the area of the minimum convex polygon created by each community in multidimensional trait space (*Villéger, Mason & Mouillot, 2008*), functional evenness represents how evenly abundance is distributed throughout multidimensional trait space (*Villéger, Mason & Mouillot, 2008*), and functional dispersion refers to the spread of species within multidimensional trait space, or their functional similarity to each other, accounting for species abundance (*Laliberté & Legendre, 2010*). We identified five morphological traits that reflect a bird's diet, flight ability, walking ability, size, and vocalization. Beak size (PC1) was taken from *Pigot et al. (2020)* to describe the birds' diet. Hand-wing index, body mass, and wing-length-corrected tarsus length were taken from the Avonet trait database (*Tobias et al., 2022*) to describe their flight ability, size, and walking ability. Peak vocalization was aggregated from several different sources (*Mikula et al., 2021*; *Hu & Cardoso, 2009*; *Francis, 2015*; *Francis & Wilkins, 2021*). Peak vocalization frequency was calculated for Eurasian collared-dove (*Streptopelia decaocto*) and Anna's hummingbird (*Calypte anna*) using Xeno-canto recordings because vocalization frequencies for these species were not included in the other sources. Three different recordings were selected for both of these

species, and up to five random calls from each recording were used to measure the peak frequency with Audacity 3.7.0 (Hamming window and fast Fourier transformation = 512) (*Francis, 2015*). Vocalization frequencies were not included for turkey vulture (*Cathartes aura*), double-crested cormorant (*Nannopterum auritum*), or great blue heron (*Ardea herodias*) because they do not vocalize as frequently as other species, and they were not observed vocalizing at any of the sites. However, they were still included in this analysis.

The relationship between canopy cover and functional evenness did not appear to be linear therefore we compared linear and second-degree polynomial models to determine which best reflected the relationship between functional evenness and canopy cover. Based on $AIC_c$, we concluded that the second-degree polynomial of canopy cover received the most support and included it in functional evenness analyses.

Values for all of the functional diversity metrics used were very small, so we rescaled them to between 0 and 100 to aid in interpretation. We used linear models to determine which environmental variables influence functional richness, evenness, and dispersion at the communities. As in previous analyses, three models were used to determine which structural, natural, and agricultural variables were useful to describe functional diversity based on $AIC_c$, *p*-values, and effect size, and these variables were included in a *post hoc* model for each functional diversity metric.

For both taxonomic and community analyses, the community at site 28 was identified as potentially having a disproportionate influence on the analysis. Site 28 was surrounded by about 78% woodland cover, 25% more than the site with the next highest woodland cover, so this could explain its high influence on the models. We explored whether its removal altered our interpretation of the models. Because it did not, we included the point in all analyses presented below.

## RESULTS

In total, we conducted 186 point counts in Edna Valley at 31 different sites, and we had 1,066 bird detections from 68 different bird species. Some of the most common species seen within the vineyards include house finch (*Haemorhous mexicanus*), lesser goldfinch (*Spinus psaltria*), song sparrow (*Melospizia melodia*), and California towhee (*Melozone crissalis*).

### Species occupancy

We found notable associations in occupancy and the various environmental variables for six different species (Fig. 2; Table S2). Sound level was most commonly influential for species occupancy. It was positively related to California quail and cliff swallow (*Petrochelidon pyrrhonota*) occupancy and negatively related to acorn woodpecker occupancy (California quail $\beta$ = 0.808, SE = 0.403, z = 2.000, *p* = 0.045; cliff swallow $\beta$ = 0.366, SE = 0.219, z = 1.670, *p* = 0.095; acorn woodpecker $\beta$ = −0.403, SE = 0.199, z = −2.020, *p* = 0.043). California quail and acorn woodpecker were also positively associated with canopy cover with California quail occupancy reaching 0.50 at about 10% canopy cover and acorn woodpecker occupancy reaching 0.60 at about 10% canopy cover (California quail $\beta$ = 0.469, SE = 0.246, z = 1.910, *p* = 0.057; acorn woodpecker $\beta$ = 0.130,

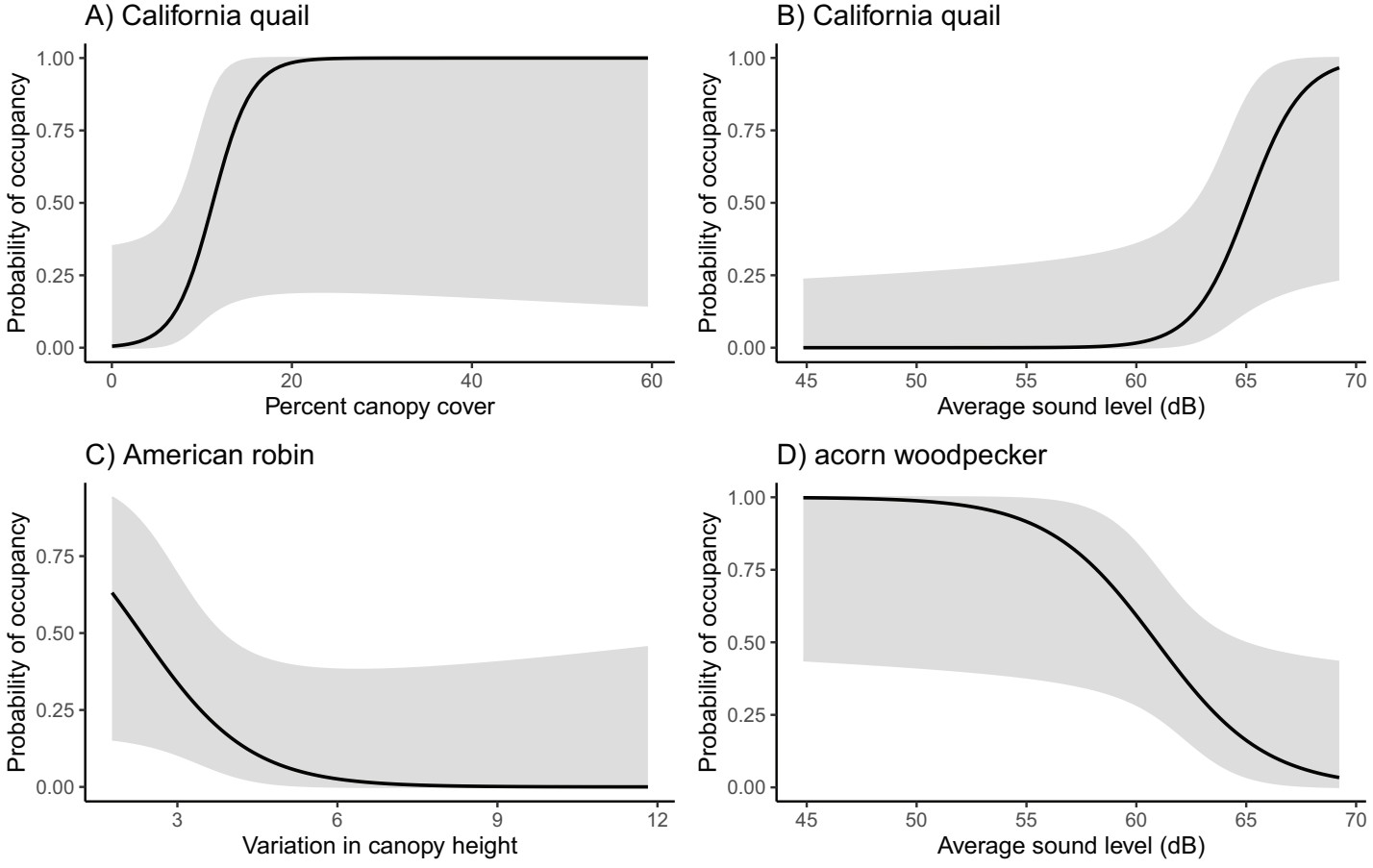

**Figure 2 Probability of occupancy in vineyards of California quail, American robin, and acorn woodpecker.** California quail probability of occupancy increases with (A) canopy cover and (B) sound level, (C) American robin occupancy decreases with increasing variation in canopy height, and (D) acorn woodpecker probability of occupancy decreases as sound levels increase.

SE = 0.064, z = 2.020, p = 0.044). The only species with a notable association with another agriculture type was the European starling which demonstrated a slight decline in occupancy with orchard cover (European starling β = −0.1226, SE = 0.067, z = −1.838, p = 0.066).

Wrentit occupancy was associated with shrubland cover, and American robin (*Turdus migratorius*) occupancy was negatively associated with canopy height variation (Wrentit β = 0.215, SE = 0.118, z = 1.830, p = 0.068; American robin β = −0.985, SE = 0.549, z = −1.795, p = 0.073). None of the species' occupancy was strongly influenced by the amount of vineyard cover. However, vineyard cover was included in the final model for several species (Table S2).

## Taxonomic community diversity

Results for coverage-standardized Shannon index and species richness were very similar, with canopy cover, vineyard cover, and shrubland cover included in the final models for both metrics (Table S3). For simplicity, we refer to both Shannon index and species

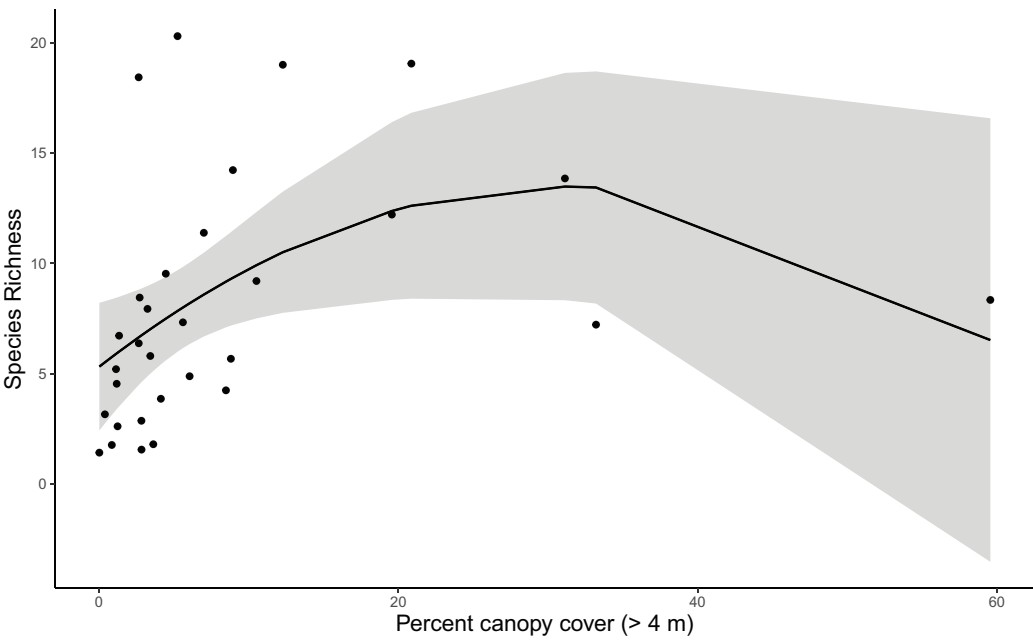

**Figure 3 Species richness increases with canopy cover until about 35% cover (species richness β = −10.786, SE = 4.946, z = −2.181, p = 0. 038; see Table S5 for other polynomial estimates).** Trend reflects marginal effects and the shaded area represents the 95% confidence interval.

richness as community diversity. Vineyard cover initially appeared informative for community diversity based on the anthropic cover model. However, when added to the *post hoc* model with canopy and shrubland cover, it was not a strong predictor for community diversity (Tables S4, S5). Canopy cover was strongly correlated with community diversity, and diversity increased with canopy cover up to about 35%, after which there were too few point count locations with higher canopy cover to produce precise estimates (Tables S4, S5) (Fig. 3).

As in the community diversity models, vineyard cover was selected for the beta diversity *post hoc* model (Table S6); however, it had no explanatory power for community composition in the *post hoc* model (Table S7). Shrubland cover was marginally useful in explaining community turnover (Shrubland $R^2$ = 0.042, F = 1.442, p = 0.171), but canopy cover and average sound level were much more informative (Canopy cover $R^2$ = 0.065, F = 2.202, p = 0.016; sound level $R^2$ = 0.050, F = 1.697, p = 0.062) (Fig. 4). Overall, the final model explained about 24% of turnover among communities.

### Functional community diversity

Based on linear models, we found that community functional evenness increased with canopy cover up to about 35% (β = −38.132, SE = 20.515, t = −1.859, p = 0.045; see Table S10 for other polynomial estimates) (Table S9; Fig. 5). Communities also had lower functional evenness when they were closer to surface water; however, the magnitude of this effect was small (surface water β = 0.063, SE = 0.031, t = 2.049, p = 0.050).

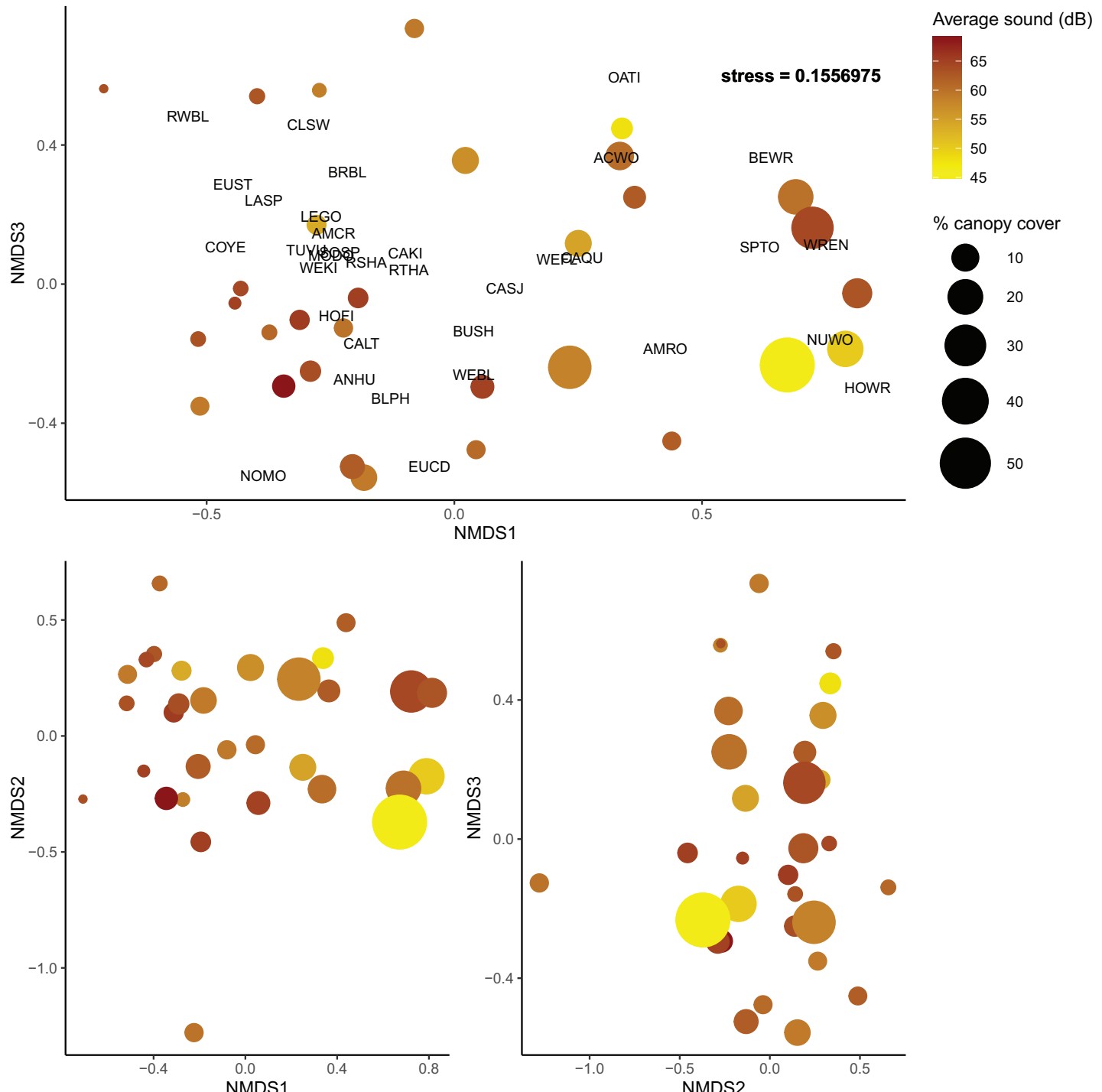

**Figure 4 Non-metric multidimensional scaling plots of bird communities in Edna Valley, CA.** Each dot is a site with the color of the dots representing the average sound level (dB), and size of the dot representing the percent canopy cover. Species detected at more than five sites are also plotted on the upper panel with species codes listed in Table S8. Three dimensions were adequate to describe variation among the communities (stress = 0.156)

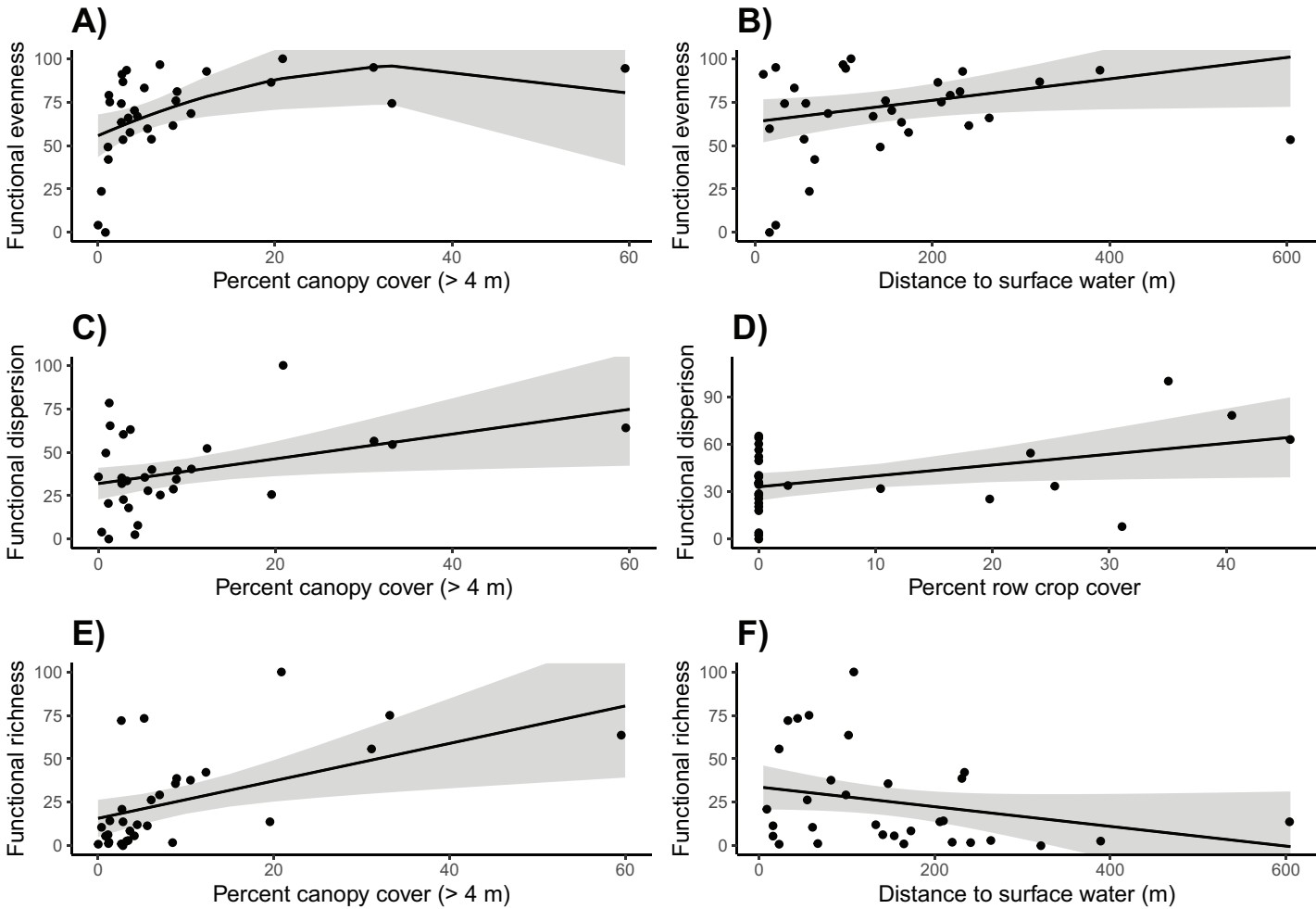

**Figure 5 Functional community diversity.** Canopy cover is correlated with higher (A) functional evenness, (C) functional dispersion, and (E) functional richness. (B) Functional evenness is positively correlated with distance to surface water, (D) Row crop cover is positively correlated with functional dispersion, and (F) functional richness is negatively correlated with the distance to surface water. Trends reflect marginal effects and the shaded areas represent 95% confidence intervals.

Functional dispersion of the bird communities was strongly and positively correlated with both row crop and canopy cover (row crop $\beta$ = 0.689, SE = 0.312, t = 2.209, $p$ = 0.036; canopy cover $\beta$ = 0.713, SE = 0.302, t = 2.360, $p$ = 0.026) (Tables S11, S12; Fig. 5). However, only nine of the 31 sites were surrounded by row crops, thus functional dispersion relationships with this variable should be interpreted with caution.

Canopy cover was again included in the final model for functional richness and had a strong positive influence ($\beta$ = 1.078, SE = 0.385, t = 2.798, $p$ = 0.009) (Tables S13, S14; Fig. 5). Distance to surface water was weakly correlated with functional richness; however, the effect was very minimal and potentially not biologically relevant ($\beta$ = −0.057, SE = 0.033, t = −1.728, $p$ = 0.095).

## DISCUSSION

Although research in the last decade has provided important insights for bird-friendly vineyard management *via* individual species trends (*Muñoz-Sáez, Kitzes & Merenlender, 2021*), nest site selection and success (*Assandri et al., 2017*), and community diversity (*Assandri et al., 2016*), most studies have been conducted outside of the United States, and in different ecosystems, so it was unclear whether results from those studies should reflect dynamics in California vineyards due to different biodiversity contexts and histories of human land use. However, similar to our findings, *Assandri et al. (2016)* also concluded that trees had a positive impact on avian community diversity in vineyards in Italy. In contrast, *Muñoz-Sáez, Kitzes & Merenlender (2021)* found that woodland-associated species were negatively correlated with vineyard cover in California vineyards; however, they did not assess the impact of ornamental or remnant trees in vineyards, which are taken into account, although not differentiated, in this study. We found that certain woodland associated species including acorn woodpecker and California quail were positively associated with canopy, some of which were ornamental trees within the vineyards. We also found evidence that surface water, surrounding row crop cover, and noise are variables explaining avian communities. Should these variables consistently explain variation in avian communities within California wine country, they could be used as tools to promote bird-friendly vineyard management. For instance, despite a willingness to buy and sometimes even pay more for wine certified as sustainable or biodiversity-friendly (*Valenzuela et al., 2022*; *Mazzocchi, Ruggeri & Corsi, 2019*; *Vecchio, 2013*), and the success of the Smithsonian's bird-friendly coffee and cocoa (*Smithsonian National Zoo & Conservation Biology Institute, 2025*), there is currently no widely used bird-friendly wine certification to our knowledge. Scientists and viticulturists should consider whether a bird-friendly wine certification could be an economically viable tool for promoting bird-friendly viticulture landscapes. *Alonso Ugaglia et al. (2021)* showed variable but positive interest among consumers towards various wine sustainability certifications. Still, much work is needed to understand the social and economic feasibility of a bird specific certification. The associations between vineyard characteristics and avian communities identified in this study could form the building blocks of these efforts.

### Species occupancy

Of the 22 species that were analyzed, only six species appeared to have occupancy patterns that were related to the environmental variables we considered (Fig. 2; Table S2). Both California quail and cliff swallows were positively associated with sound level, and acorn woodpeckers were negatively associated with sound. These different responses may reflect underlying differences in sensitivity to anthropogenic conditions. Acorn woodpeckers and California quail were also positively associated with increasing canopy cover which aligns with their woodland and scrub habitat preferences. Similarly, wrentits were more likely to occupy vineyards with more shrubland cover corresponding to their known habitat preference for shrubland. American robins were negatively associated

with canopy height variation, which likely reflects their preference for environments with tall trees.

Surprisingly, the amount of vineyard cover was not strongly correlated with the occupancy of any species. This could indicate that vineyards are tolerable for the species we analyzed, and that certain management interventions such as increasing canopy cover could make vineyards of varying sizes habitable for birds.

## Taxonomic community diversity

Our taxonomic diversity analyses showed clearly that canopy cover is the most important predictor for avian community diversity in and around Edna Valley vineyards. Community diversity increases sharply as canopy cover increases up to about 35%, indicating that even small increases in canopy within vineyards could attract a diverse community of birds. This could be caused by the increase in microhabitats available to birds, or the presence of these trees could be a valuable source of food, shelter, and nesting substrates or materials for the birds. Other research in agricultural and urban areas have also found that tree cover can increase avian diversity (*Assandri et al., 2016*; *Barth, FitzGibbon & Wilson, 2015*; *Wood & Esaian, 2020*). Taxonomic diversity appears to decrease above 35% canopy cover, although the precision of taxonomic diversity estimates with more canopy was low because we only sampled three communities in areas with high canopy cover. This pattern reflects that most current vineyards have relatively low canopy cover. However, it is possible that taxonomic diversity does actually decrease with higher percent canopy cover because the habitat becomes less attractive to grassland and shrubland birds. To fully evaluate whether high levels of canopy cover truly is associated with declines in taxonomic diversity, future research could target vineyards that have or are surrounded by a high proportion of canopy cover. Also needed are studies that explicitly test whether the benefits of tree cover for avian diversity might depend on whether trees are native, naturally occurring trees or if they are non-native ornamental trees.

We also found that community turnover was influenced by anthropic and structural characteristics of the sites. Canopy cover appears to influence the community composition of these bird communities as well as the overall diversity, and sound is only associated with a shift in the community composition. Quiet sites with high canopy cover had bird communities that were very different from loud sites with low canopy cover. This could reflect the differences in urban tolerant and urban intolerant bird communities that have been noted in other studies, as some birds are less susceptible to interference from loud noise (*Francis, 2015*), such as the cliff swallow and California quail in this study (Table S2). The community turnover associated with variation in canopy cover could be a reflection of a transition from grassland associated birds that prefer lower canopy cover to woodland associated birds that prefer higher canopy cover. Previous research supports this finding as canopy cover can increase the abundance of woodland birds in human-modified environments (*Bennett et al., 2014*). However, as discussed previously, canopy cover was also associated with higher overall taxonomic diversity compared to communities with low canopy cover.

## Functional community diversity

The percent canopy cover at each site was overwhelmingly informative for predicting the functional diversity of the avian community. Canopy cover was positively correlated with functional evenness, dispersion, and richness. In other words, increases in canopy cover is not only associated with more ecological function, it also promotes a more even distribution of birds across functional space and is associated with more dissimilar functions. This finding suggests that even small quantities of trees serve as a keystone structure for birds and helps support a diverse and resilient avian community in vineyards and even a small number of trees appear to have a similar benefit to avian diversity in other human-modified landscapes. For example, scattered trees are associated with high bird diversity and abundance in livestock grazing fields, urban parks, and conifer plantations (*Fischer, Stott & Law, 2010*; *Stagoll et al., 2012*; *Yamaura, Unno & Royle, 2023*). This can in turn increase the services that the avian community can provide for vineyards including insect and rodent management and tourism opportunities (*Barbaro et al., 2014*; *Mols & Visser, 2002*), which become more important as agricultural lands expand.

Our results indicate that variation in crop type cover generally increases the diversity and evenness of avian communities in vineyards. Row crop cover was positively associated with functional dispersion indicating that row crops attract different birds than vineyards do. These birds include red-winged blackbirds (*Agelaius phoeniceus*) and Brewer's blackbirds (*Euphagus cyanocephalus*), which were almost exclusively found at vineyards with oats growing nearby. Incorporating other crops such as oats or strawberries into vineyards could benefit vineyard owners by allowing them to maintain or even increase crop productivity while sharing the land with a diverse avian community. Understanding whether the row crop type contributes to taxonomic and functional diversity is necessary to confirm whether all row crops positively influence avian communities.

Community functional diversity is also influenced by proximity to surface water and the proportion of surrounding land covered by vineyards. Nearby natural and human made streams and ponds increase the trait space that avian communities in vineyards occupy. Based on observations during the surveys, the increase in functional richness was likely driven by mallards (*Anas platyrhynchos*) and American coots (*Fulica americana*), which were commonly detected in ponds. Woodland birds could also be contributing to the increased richness near streams as riparian vegetation has proven to be very beneficial for woodland birds in agricultural areas (*Jedlicka, Greenberg & Raimondi, 2014*). Our results also reveal that communities at sites with the highest proportion of vineyard cover had lower functional evenness indicating that large concentrations of vineyards may create an unbalanced distribution of functions within the community. That is, communities embedded within high vineyard cover are dominated by common species with few rare species. This can result in a vulnerable community where the loss of one of these rare species could cause a large difference in ecosystem functions that the bird community provides.

Overall, sites with high taxonomic diversity also had high functional richness (Fig. 6). However functional richness does not take into account the spread of species in trait space,

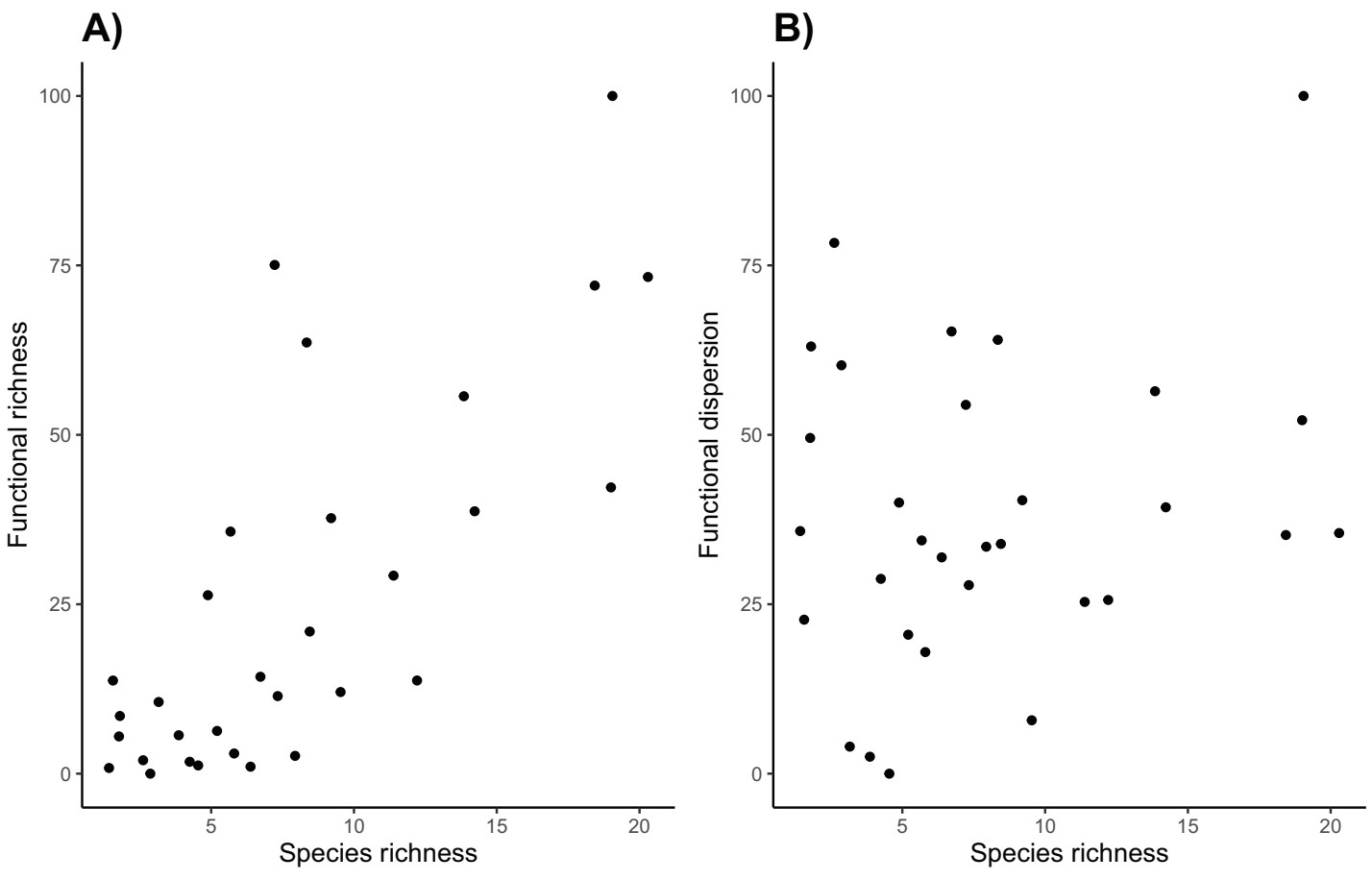

**Figure 6 Taxonomic and functional diversity.** Species richness is (A) positively correlated with functional richness ($R^2$ = 0.573, $p < 0.001$), but (B) it is not correlated with functional dispersion ($R^2$ = 0.053, $p = 0.215$).

and functional dispersion of these communities is not strongly correlated with taxonomic diversity. While high canopy cover is associated with high functional dispersion and taxonomic diversity, many sites with higher species richness do not have particularly high functional dispersion. This is perhaps because many of the species have similar traits, so they do not increase the functional dispersion of the community, highlighting the importance of using both functional and taxonomic diversity to assess avian communities (*Mazel et al. 2018*). Vineyard managers are likely most interested in functional diversity of the avian community on their land, as it can be closely tied to ecosystem functions that the birds can provide (*Barbaro et al., 2014*). Specifically, increasing the presence of functionally distinct birds such as raptors, and insectivorous songbirds could help manage a host of agricultural pests (*Kross, Bourbour & Martinico, 2016*; *García, Miñarro & Martínez-Sastre, 2018*).

## CONCLUSION

Increasing canopy cover up to about 35% in viticultural landscapes appears overwhelmingly to be the most effective way to increase avian community diversity;

however, adding land cover and structural variation in general may also help. Future research focused on the impact of different row crop types or species of trees in vineyards would be extremely valuable. Investigation into viticulture practices, such as pesticide use, to determine if a reduction in prey availability may negate the benefits of canopy cover and land cover variation will also be helpful for developing more bird-friendly management recommendations. Loss of habitat from agriculture is a major contributor to bird population decline in North America (*Stanton, Morrissey & Clark, 2018*), therefore it is vital that we encourage diverse and resilient bird communities within vineyards and take advantage of natural pest management that birds can provide.

### Funding
This work was supported by the Sea and Sage Audubon Society and the California Polytechnic State University, San Luis Obispo Biological Sciences Department. The funders had no role in study design, data collection and analysis, decision to publish, or preparation of the manuscript.

### Grant Disclosures
The following grant information was disclosed by the authors:
Sea and Sage Audubon Society and the California Polytechnic State University, San Luis Obispo Biological Sciences Department.

### Competing Interests
The authors declare that they have no competing interests.

### Author Contributions
- Lindsay Peria conceived and designed the experiments, performed the experiments, analyzed the data, prepared figures and/or tables, authored or reviewed drafts of the article, and approved the final draft.
- Clinton D. Francis conceived and designed the experiments, analyzed the data, authored or reviewed drafts of the article, and approved the final draft.

### Data Availability
Raw data is available at Zenodo:
Peria, L., Francis, C. D. (2025) Avian-biodiversity-in-vineyards: Data and code for "Avian biodiversity in central California vineyards" (v1.1.0). Zenodo. https://doi.org/10.5281/zenodo.15307110.

### Supplemental Information
Supplemental information for this article can be found online at http://dx.doi.org/10.7717/peerj.19904#supplemental-information.

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
