# Peer review of "Avian biodiversity in central California vineyards"

_PeerJ, doi:10.7717/peerj.19904_

## Round 0.1 · original submission · Minor Revisions

Thank you very much for your manuscript titled “Avian biodiversity in central California vineyards” that you sent to PeerJ.

This study presents very valuable and relevant information on taxonomic and functional bird diversity in vineyards from California.

As you will see below, comments from referee 1 and 2 suggest a minor revision before your paper can be published. Given this, I would like to see a minor revision dealing with the comments. Their comments should provide a clear idea for you to review, hopefully improving the clarity and rigor of the presentation of your work. I will be happy to accept your article pending further revisions, detailed by the referees.

Reviewer 1 suggests adding information on the taxonomic and functional diversity of birds, as well as including in the discussion the impacts of biodiversity conservation on vineyard birds. He also makes some grammatical observations.

Reviewer 2 suggests adding more details to the information provided, in order to facilitate the reading and understanding of the study.

Please note that we consider these revisions to be important and your revised manuscript will likely need to be revised again.

·

Basic reporting

The paper has a clear focus on avian biodiversity in Californian vineyards. The results are broken down nicely into how vineyard characteristics affect species occupancy, taxonomic diversity, and functional diversity. It is well written and seems technically sound; I have no critique of the methodological details. I believe that small improvements could be made, primarily by introducing taxonomic and functional diversity, and expanding on the benefits of increased avian diversity.

Good background information is given on both agriculture in a broad sense and vineyards in California in a specific sense. It is clear how this research fits into the field of knowledge.

• The main addition to the introduction I would suggest is background information on taxonomic and functional diversity, given their importance to the research. What are the benefits to increasing these measures of diversity?
• Another potential addition could be briefly discussing sparing vs sharing biodiversity impacts. Is land sparing appropriate in vineyards? I would suggest land sharing may be more feasible.

• I have a few small grammatical suggestions: Use hyphens for compound adjectives throughout, e.g. “wildlife-friendly practices”. Line 27: use “,” instead of “;”. Change tense from increases to increased (line 288). Space needed after brackets (line 395).

The structure of the paper is very good with clear subheadings making it easy to follow. Species occupancy is however not given the same status in the discussion as it was in the methods and results. The figures are all relevant and look very good overall.

• I suggest expanding your discussion on the species occupancy results, ideally with its own subheading to be consistent with the rest of the paper.
• Figures 2 and 6 should be referenced in the results.
• Figure 4 legends could be tidier. There is part of a larger dot present in the % canopy cover legend which can be removed. The other legend does not have a title, I suggest adding average sound level (dB). Some dots look darker than the scale, check that the scale includes the full dot colour range.

Experimental design

The experimental design is thorough and enables the hypotheses to be answered. The methods are described in good detail.

• The point counts are conducted in just over seven weeks. I suggest briefly mentioning whether there is a seasonal effect to your results.
• Can the USGS national hydrography dataset be referenced?

Validity of the findings

The findings are clearly stated and relate well to the aims of the research.

• In figure 3, the trend mentioned of a decrease in species richness after 20% canopy cover seems to be driven by just two data points. I would be careful discussing this as a decreasing trend.
• In figure 5a there is a non-linear relationship and I don’t believe a linear line is appropriate. There is a sharp increase in functional evenness up to 10/15% canopy cover, but then there is a plateau. This could be analysed as a gam or linear/quadratic plateau model. Figure 5f is also potentially non-linear.

Additional comments

To increase the impact of the research, I would suggest expanding on the benefits of increased avian biodiversity. What are the main aims behind increasing avian diversity in vineyards? More biodiversity is good, and as biologists we both say and hear this frequently, but what are the specific benefits? Is any kind of increase in avian biodiversity as valuable as any other increase? For example, if all vineyards install a small pond and increase their avian diversity by adding a pair of mallards, is that as beneficial as providing habitat for and attracting an endangered species? This could be linked to functional diversity. This is briefly discussed at the end of the discussion (line 412) but should be a more important part of the discussion. People, such as vineyard owners, often like to hear about direct benefits to them. Ecosystem services are mentioned, but greater detail would be useful. Do vineyard owners have any extrinsic incentive to increase avian diversity e.g. financial incentives? This is mentioned in the discussion (line 331) but could also be included in the introduction.

·

Basic reporting

Pass

Experimental design

Pass.

Validity of the findings

Pass.

Additional comments

In this study the authors investigated the effects of grassland, shrubland, vineyard, orchard, row crop, and developed land cover, canopy cover, canopy height, variation in canopy height, and distance to surface water on avian species occupancy, community diversity, and functional diversity in California vineyards. I congratulate the authors on an excellently written manuscript with respect to grammar. It was a pleasure to read. I am particularly fond of the community turnover analyses. Most of my comments are relatively minor and are focused on providing some additional detail to help the reader better understand the study system.

Major comments:

Line 51: Canopy has to be explicitly defined. For example, the reader only knows that it is something > 4m, but what actually is it referring to? Tree cover, etc.?

Lines 202-204: Linear models are fine for diversity, but richness is a count, therefore, assumption of the linear model are violated as variance always increases with counts. Please use the appropriate Poisson glm.

Statistical models in general: how did you evaluate model fit and did you check for collinearity or correlation among your predictors? If not, please consider doing this and if so, please provide text in the methods describing what you did. I am not big on p-values in general – I am more of an effect size person – but please state how you interpreted support for statistical effects in your methods, given that you report on a lot of marginally in-significant trends (beginning on line 271).

Lines 408-409: This last paragraph is challenging to understand because the spread of species in multidimensional trait space is lacking biological relevance in the methods (line 231). I would let the reader know in the methods what functional dispersion is actually capturing and then in the discussion tell the reader why this result is important. I wouldn’t assume that many readers know what functional dispersion refers to.

Minor comments:

Line 36: I would also cite something more recent about drivers of global biodiversity decline.

Line 44: Please define the term viticulture the first time you use it for readers not familiar with this term.

Lines 57-59: please re-word this sentence as the same thing is stated twice – bird diversity is lower and communities are less diverse. If you are referring to the ecological community in general, please be more specific. As it currently reads, it could be interpreted as referring to the avian community twice.

Lines 67-68: please strengthen argument as to why California is different or why a study is needed in California relative to the geographies where the other studies occurred. E.g., Avian community composition differs and therefore it is possible that bird species respond differently in California compared with other areas.

Line 88: I am not sure you measured landscape features as opposed to patch level characteristics or local land cover characteristics. I would use different terminology. Simple fix. Landscape structure is likely to influence what you measured given larger and smaller source populations/spill-over, connectivity, etc.

Line 124: Please describe what a faux fur widescreen is.

Line 172: please describe what models were fit for each hypothesis – I am assuming all subsets. Please also describe why you chose the modelling approach you chose with three subsets of models as opposed to fitting a global model and fitting all subsets.

Line 184: please describe what threshold was used for sufficient detection histories. Please also explain why occupancy models instead of binomial models, controlling for the detection variables. What analytical advantage did they provide if any?
Line 195: please provide evidence that number of species observed did not approach actual number of species at each site or consider rephrasing.

Line 196: please define survey completeness or coverage.

Lines 214-215: I suggest editing this text to better describe what was done. For example, it reads as if you only looked at percent canopy cover, but if you read below these lines, it reads as if you analyzed all the variables that you used in your diversity models.

Lines 236-238: I would just explain that these two species were not included in the other sources. Otherwise, one might speculate that they may have been there but ignored or were erroneous, etc. In other words, why did you have to use Xeno-canto for these two species.

Lines 245-248: Not necessary to do, but you could have standardized by subtracting the mean and dividing by the SD. I generally do this for all continuous predictors when possible, regardless of the study, if interested in relative effect sizes.

Line 256: I suggest a bit more detail here. E.g., interpretation in terms of effect sizes or statistical significance, or both.

Line 264: Note that your beta parameter isn’t showing up in the PDF version I downloaded.

Lines 275-276: Wrentit occupancy – why was it strongly associated with shrubland cover? What did you use to define the strength of association? The effects size was relatively small at 0.216 and the p-value was 0.068. Was it related to the AICc value? Please consider changing wording. Similarly, why is 0.462 a considered a small effect size on line 303. This seams relatively large.

Line 289: I would argue that it continues to increase to about 35% before declining based on the curve you fit to the data in figure 3. So the decrease isn’t after 20%. Please also refer to line 349.

Line 325: please explain how ornamental trees were taken into consideration in this study. It isn’t clear from the methods.

Lines 337-339: there is a whole literature on this topic. I suggest providing a couple citations as examples when discussing feasibility.

Line 354: I would speculate on why diversity decreases after 35% canopy cover if this is a real pattern. This is low hanging fruit given you analyzed the entire avian community.

Lines 381-388: Based on the results reported, it is hard to understand how, based on the results presented, land cover heterogeneity resulted in increased diversity and evenness. I am not sure land cover heterogeneity was fully captured in this study. For example, in landscape ecology, this generally refers to the diversity of land cover types. Also – the results about the oats came out of nowhere for the reader because this level of details isn’t provided in the methods or results.

Line 395: space needed between the closing parenthesis and “which”.

Lines 399-402: This line is describing community richness, but not functional evenness. Also, doesn’t low functional evenness suggest that rare functional traits are present? In other words, I might expect high functional evenness among a few traits as land cover becomes more homogeneous.

Lines 416-418: I would re-word to reflect the non-linear relationship you found, otherwise, as written, it suggests increasing canopy cover beyond 35%.

Line 420 – Please be more explicit about how pesticide use may negate the benefits of canopy cover to birds (?). Is it because the invertebrate community is declining or are birds being poisoned, etc.?

---

## Round 0.2 · accepted · Accept

After reviewing this revised version of your manuscript, I see that the main comments suggested by the reviewers have been included, while the suggestions not considered are justified in detail. Therefore, I am satisfied with the current version and consider it ready for publication.

·

Basic reporting

I am satisfied with the basic reporting.

Experimental design

I am satisfied with the experimental design.

Validity of the findings

I am satisfied with the validity of the findings.

Additional comments

I think the revisions have improved the manuscript and any issues have been addressed.